# Tetraethylenepentamine-Grafted Amino Terephthalic Acid-Modified Activated Carbon as a Novel Adsorbent for Efficient Removal of Toxic Pb(II) from Water

**DOI:** 10.3390/molecules29071586

**Published:** 2024-04-02

**Authors:** Mutairah S. Alshammari

**Affiliations:** Chemistry Department, College of Science, Jouf University, P.O. Box 2014, Sakaka 72341, Saudi Arabia; msshamari@ju.edu.sa; Tel.: +966-504904183

**Keywords:** tetraethylenepentamine (TEPA), adsorption, toxic Pb(II), isotherm

## Abstract

In this study, a new composite, tetraethylenepentamine (TEPA), was incorporated into amino terephthalic acid-modified activated carbon (ATA@AC) through a one-pot integration of TEPA with the COOH moiety of ATA@AC. This process resulted in the creation of a TEPA@ATA@AC composite for Pb(II) removal from an aquatic environment. Several techniques, including SEM, EDX, FT-IR, TGA, XRD, and Zeta potential, were employed to emphasize the chemical composition, morphology, and thermal durability of the as-synthesized TEPA@ATA@AC composite. The impact of experimental variables on the adsorption of Pb(II) ions was studied using batch adsorption. The uptake assessment suggested that the TEPA@ATA@AC composite exhibited superior Pb(II) removal performance with high removal efficiency (97.65%) at pH = 6.5, dosage = 0.02 g, equilibrium time = 300 min, and temperature = 298 K. The isotherm data exhibited good conformity with the Langmuir isotherm model, whereas the kinetics data displayed strong agreement with both pseudo-first-order and pseudo-second-order kinetics models. This reflected that the Pb((II) uptake by the TEPA@ATA@AC composite was caused by physisorption coupled with limited chemisorption. The greatest monolayer uptake capacity of the TEPA@ATA@AC composite was 432.8 mg/g. The thermodynamic findings indicated that the Pb(II) uptake on the TEPA@ATA@AC composite was an exothermic and feasible process. After five adsorption—desorption runs, the TEPA@ATA@AC composite maintained a superior uptake capacity (83.80%). In summary, the TEPA@ATA@AC composite shows promise as a potent adsorbent for effectively removing Cr(VI) from contaminated water, with impressive removal efficiency.

## 1. Introduction

Water contamination with hazardous elements is a global problem, threatening both aquatic ecosystems and human health owing to their toxicity and non-degradable, bioaccumulative nature [1,2,3]. Major industries, including the chemical industry, paint, tanning, mining, textile, storage battery, plastics, pigment, and mining industries have played a role in the wide distribution of hazardous elements in the water environment [4,5]. One of the poison contaminant elements in the water environment is lead [Pb(II)], which is a toxic metal, carcinogenic, and harmful to aquatic ecosystems and humans [6,7]. Wastewater from battery manufacturing, electroplating, dye, and paint has been found to contain high content of Pb(II) ions, exceeding the World Health Organization (WHO) standards (0.01 mg/L) [8]. Therefore, effectively removing it from wastewater is both important and challenging. The primary techniques employed for eliminating Pb(II) ions from wastewater encompass ultrafiltration membrane, electrochemical, ion exchange, solvent extraction, reverse osmosis, coagulation, bio-removal, and uptake processes [2,9,10,11]. Among these approaches, the adsorption process is a significant approach for heavy metal uptake owing to its cost-effectiveness, facile operation, high removal efficacy, environmental friendliness, and ability to regenerate the adsorbent material [12,13,14,15]. Different adsorption materials have been applied, including pure materials, raw biomass, natural and synthetic, inorganic, organic, and composite materials for the uptake of Pb(II) and other ions. The utilization of activated carbon derived from low-cost materials has garnered considerable attention in diverse environmental applications, particularly in removing heavy metals. This interest stems from its non-toxic nature, high specific surface area, stable structure, and excellent removal performance. With its abundant active surface and modified moieties, activated carbon demonstrates a significant affinity for heavy metals. These functional groups greatly influence the surface characteristics of carbon-based materials, facilitating the binding of hazardous substances to them [16].

To improve the uptake capability of hazardous elements, the surface features of activated carbon were boosted by modifying its surface with different materials containing functional groups such as thioglycolic acid [17], 2-((2-aminoethylamino)methyl)phenol [18], H_2_O_2_ [19], chitosan/poly(ethylene oxide) [18], sodium thiosulfate [20], chitosan/montmorillonite clay [21], Ag-SiO_2_ [22], and PEI/chitosan [23]. Amino-modified activated carbon has been used to eliminate diverse contaminants such as heavy metal ions and anions [24], CO_2_ [25], and organic pollutants [24]. Li et al. prepared mesoporous-activated carbon from sugarcane bagasse and modified it with nitric acid and ethylenediamine to produce amino-modified MC I7-EDA. The MC I7-EDA was employed for the uptake of Pb(II) ions from aqueous solutions. They found that the amino groups grafted onto porous activated carbon enhanced the Pb(II) adsorption compared to unmodified activated carbon (MC I7). The greatest Pb(II) adsorption capacity of MC I7-EDA reached 150 mg/g, which was approximately 1.5 times greater that of MC I7 at 200 mg/L. This was attributed to the amino groups introduced on the surface of the mesoporous activated carbon [26]. Amine-modified corn husk-derived activated carbon was prepared by Ismail et al. for the elimination of Ni(II), Cu(II), and Pb (II) from battery recycling contaminated water. They found that the highest uptake capacity of Ni(II), Cu(II), and Pb(II) ions was 0.337 0.724, and 2.814 mg/g, respectively [27]. Li et al. fabricated SH-modified activated carbon from sewage sludge mixed with coal for hazardous elements uptake from water. The adsorbent material demonstrated efficacy in the uptake of nickel, cadmium, lead, and copper, with uptake capacities of 52, 87, 96, and 238 mg/g, respectively [17]. Aminosilane-modified activated carbon (AFAC) was prepared by Ha et al. for the uptake of Cd, Ni, and Zn from aqueous media. The maximum adsorption capacities, as determined using the Langmuir equation, were 204.3, 242.5, and 226.9 mg/g for Zn(II), Cd(II), and Ni(II), respectively [28].

Tetraethylenepentamine (TEPA: C_8_H_23_N_5_) is a polyamine containing five amino groups and a high surface functionality. TEPA is utilized as a ligand recommended for extra Cu(II) removal in patients with Wilson’s disease [29]. It is used to produce adsorbents for the uptake of toxic ions [30], dyes [31], and CO_2_ owing to the plentiful nitrogen atoms in its composition, making it better suited for applications in adsorption and biomedicine [30,32,33,34,35,36]. Numerous recent studies have proven that TEPA can boost the adsorption characteristics of different adsorbent materials [30,32,33,34,35,36]. Tang et al. prepared amino and thiol-functionalized AC for lead and cadmium removal from water. They found the highest adsorption uptakes of 142.03 and 279.20 mg/g for lead and cadmium, respectively, using amino-activated carbon (N-AC), and 232.02 and 130.05 mg/g for Pb(II) and Cd(II), respectively, using thiol-activated carbon (S-AC) [36]. Xu et al. prepared TEPA-GO@MnFe_2_O_4_ for Pb(II) removal. They found that TEPA-GO@MnFe_2_O_4_ exhibited high uptake efficacy for Pb(II) (263 mg/g) compared to GO@MnFe_2_O_4_ (133 mg/g) and pristine GO (196 mg/g) [30].

This work introduces a novel, facile, low-cost, and effective approach for preparing of tetraethylenepentamine-grafted amino terephthalic acid-modified activated carbon (TEPA@ATA@AC). The objective was to evaluate its applicability for the adsorptive removal of Pb(II) ions from aqueous solutions. The prepared composite was characterized by numerous amino groups on its surface, imparting a high affinity for Pb(II) adsorption. To the best of our knowledge, this study represents the first investigation into synthesizing of the TEPA@ATA@AC composite as a new adsorbent for the uptake of Pb(II) ions from the aquatic environment. This study explores the impact of the duration time, initial Pb ion concentration, pH value, and adsorbent dosage through batch adsorption experiments. Additionally, adsorption properties, including isotherms, kinetics, thermodynamics, and reusability, are thoroughly evaluated and discussed.

## 2. Results and Discussion

### 2.1. Characterization of Adsorbent

The FTIR spectra of AC (a), TEPA@ATA@AC (b), TEPA (c), and Pb(II)-loaded TEPA@ATA@AC (d) are illustrated in Figure 1a. The FTIR spectra of AC (line a) showed a broad band at 3442 cm^−1^, indicating the presence of –OH stretching of phenolic, alcohol, and carboxylic acid compounds. There were bands at 2844–2914 cm^−1^ (C-H aliphatic stretching), 1733 (-COO- stretching), 1635 cm^−1^ (aromatic -C=C stretching vibrations), and 1377–1464 cm^−1^ (C-H aliphatic bending). The bands at 1071 cm^−1^ and 831 cm^−1^ were attributed to the ν(C-O bond) [37,38,39]. For line c, the characteristic peaks of TEPA could be ascribed as follows: 3183 and 3352 cm^−1^ (–NH_2_ stretch), 2925 and 2811 cm^−1^ (–CH asymmetric and symmetric stretch), 1606 cm^−1^ (–NH_2_ band), 1617 cm^−1^ (–NH_2_ bend), 1461 and 1377 cm^−1^ (–NH bend), 1119, 1054, and 1034 cm^−1^ (C–N stretch) [40,41]. The FTIR spectra of TEPA@ATA@AC (line b) showed bands at 3441 cm^−1^ (–OH and –NH_2_ stretch), 2925 and 2850 cm^−1^ (– CH asymmetric and symmetric stretch), 1654 cm^−1^ (-NHCO- amide band), 1681 cm^−1^ (–NH_2_ bend or aromatic -C=C stretching vibrations), 1477 cm^−1^ (–NH bend), 1417 cm^−1^ (-COO- stretch), 1110 cm^−1^ (C-N stretch), and 1050 cm^−1^ (C-O-C stretch). The appearance of characteristic bands for amide (-NHCO), -NH, and C-N bonds indicated that TEPA successfully interacted with the COOH groups on the ATA@AC surface. All these findings confirmed the development of the TEPA@ATA@AC composite. After the uptake of Pb(II) (line d), the intensity of some characteristic bonds declined or disappeared due to the binding of Pb(II). In detail, the peaks at 3341 cm^−1^ and (1581, 1471 cm^−1^) for –OH/–NH stretching and –NH_2_ bending, respectively, were slightly reduced in intensity and moved to a slightly higher wavenumber of 3345 cm^−1^ and (1583, 1474 cm^−1^), respectively, indicating the binding of Pb(II) with functional groups on the TEPA@ATA@AC composite through an electrostatic interaction. The bands of C-N and C-O disappeared after Pb(II) uptake. All the above changes indicate that the primary or secondary amino groups of the TEPA@ATA@AC contributed to the uptake of Pb(II).

The XRD patterns of the AC, ATA@AC, and TEPA@ATA@AC composite are explored in Figure 1b. For the XRD spectrum of AC, the two characteristic diffraction peaks at 2θ = 24.62° and 43.71° corresponded to the (002) and (100) planes of the semi-crystalline structure of activated carbon, respectively [41,42]. The XRD spectra of the AC functionalized with ATA/ETPA did not exhibit significant changes compared to pure AC. The diffraction angle at 24.62° showed minor shifts to lower Bragg’s angle (23.53°) after modification of AC with ATA and subsequent amination with TEPA, indicating that the functionalization of AC with ATA and TEPA does not alter the crystal architecture of AC.

The nitrogen isotherm for TEPA@ATA@AC is illustrated in Figure 2. According to the IUPAC classification, the isotherm exhibited a Type IV pattern, accompanied by a H4 hysteresis loop. Capillary condensation occurred within the pores at a relative pressure (p/po) exceeding 0.5, confirming the presence of a substantial quantity of mesoporous. The TEPA@ATA@AC material possessed a surface area, pore volume, and pore radius of 322.4 m^2^/g, 0.2731 cm^3^/g, and 3.452 nm, respectively.

The morphological structure and elemental composition of the TEPA@ATA@AC composite before and after Pb(II) uptake are illustrated in Figure 3a,b, and Table 1, respectively. It was observed that the TEPA@ATA@AC composite had an irregular shape with roughened surfaces (Figure 3a,b). Furthermore, the surface of the TEPA@ATA@AC exhibited an aggregation of activated carbon particles with organic molecules, suggesting the modification of the ATA@AC surface with TEPA molecules. The EDX analysis detected the presence of carbon (72.33%), oxygen (24.20%), and nitrogen (3.47%) elements in the structure of the TEPA@ATA@AC composite (Table 1). After Pb(II) adsorption, the surface of TEPA@ATA@AC was coated with Pb(II), indicating an interaction between the Pb(II) ions and the modification moieties of amine and hydroxyl at the surface of the TEPA@ATA@AC composite (Figure 3c). This observation was supported by the EDX investigation (Table 1). The EDX analysis showed the existence of Pb(II) in the TEPA@ATA@AC composite, with a content of 0.88%, indicating the fruitful uptake of Pb(II) onto the TEPA@ATA@AC nanocomposite (Table 1).

Figure 4a depicts the TGA/DTA curve analysis of the TEPA@ATA@AC composite. Three stages of decomposition occurred, with an initial weight loss of ~1% within the range temperature 30–180 °C owing to the desorption of water from the surface of the adsorbent. The second degradation stage started from 230 to 450 °C with weight loss of ~46%, which was related to the degradation of cellulose, hemicellulose molecules, and organic compounds of ATA and TEPA. The third weight loss of ~8% was due to the complete disintegration of the architecture of the organic compound on the surface of the AC nanocomposite.

The Zeta potential is a crucial variable for the estimation of the surface charge of a composite material. Figure 4b shows the Zeta potential with the pH. The isoelectric point (IEP) of the TEPA@ATA@AC composite was 5.82, which indicated that the surface of the TEPA@ATA@AC composite had a dual charge (positive and negative) at pH < 5.82 and pH > 5.82, respectively. Elwakeel et al. analyzed the Zeta potential of chitosan@TEPA@AC and found that the Zeta potential was close to 5.9 [43]. In a similar vein, Elwakeel observed that the Zeta potential of chitosan/amino (TEPA) resin (R1) and chitosan containing both amine and quaternary ammonium chloride moieties (R2) was 6.2 and 9.7, respectively [44].

### 2.2. Optimization Conditions

#### 2.2.1. Selectivity Study

The ability of the prepared AC, ATA@AC, and TEPA@ATA@AC composites was explored for the uptake of hazardous elements such as Cd(II), Pb(II), and Cr(III) from an aquatic environment, as shown in Figure 5a. This measurement was performed under the following conditions: ((metal): 20 mg/L; time: 24 h; adsorbent dose: 0.02 g; volume: 50 mL; temperature: 298 K; pH: 6). The results revealed that the TEPA@ATA@AC showed a higher removal efficiency for Cd(II), Pb(II), and Cr(III) metals in the following rank: Pb(II) (95.15%) > Cd(II) (88.95%) > Cr(III) (83.20%) compared to the AC and ATA@AC composite, which had the following adsorption orders: Pb(II) (84.50%) > Cd(II) (83%) > Cr(III) (77.80%) for AC and Pb(II) (86.10%) > Cd(II) (83.85%) > Cr(III) (78.50%) for the ATA@AC composite. This was attributed to the structure of the TEPA@ATA@AC composite containing multifunctional groups (NH_2_, OH, and COOH) compared to the AC and ATA@AC composite, which contained only COOH and OH groups. In addition, all the adsorbents had the greatest performance of Pb(II) uptake compared to the other ions. This was attributed to their electronegativity, hydration energies, and hydrated radii, which are the key to determining their selective adsorption [8]. A higher adsorption rate was observed for metal ions with smaller ionic radii and lower hydration energies and metal ions with greater electronegativity. The electronegativity, ionic radius, and hydration energies of Pb(II), Cd(II), and Cr(III) are 2.33, 1.69, and 1.66 Pauling; 1.20, 0.97, and 0.62 A; and −1481, −1807, and −4563 kJ/mol, respectively [45]. According to these values, the order of heavy metals adsorption on all adsorbents was Pb(II) > Cd(II) > Cr(III), confirming that the uptake process was based on the properties of heavy metals. Given that the TEPA@ATA@AC composite exhibited the highest elimination efficacy for Pb(II) at 95.15%, it was selected for detailed Pb(II) adsorption experiments in this investigation.

#### 2.2.2. Impact of Initial pH

The influence of the original pH on the uptake of Pb^2+^ ions onto the TEPA@ATA@AC composite was emphasized at the pH values between 2.6 and 7.0 at ([Pb(II)]_0_ = 20 mg/L, m = 0.02 g, T = 298 K, time = 24 h, agitation = 100 rpm), as shown in Figure 5b. The percentage of removal of Pb(II) and removal capability of TEPA@ATA@AC improved from 3.95% and 1.98 mg/g to 93.95% and 46.97 mg/g as the pH value changed from 2.6 to 6.5. Subsequently, it was slightly raised from 93.95% and 46.97 to 97% and 48.5 mg/g, respectively. The alteration of Zeta potentials of the TEPA@ATA@AC composite with the pH is plotted in Figure 4b. The Zeta potential (pH_ZPC_) of the TEPA@ATA@AC composite was 5.8. This implies that the surface of TEPA@ATA@AC had a positive and negative charge at pH < 5.82 and pH > 5.82, respectively. Similar results were observed for the Zeta potential values for chitosan- and triethylenediamine-activated carbon (Chit-TEPA@AC) [43] and magnetic chitosan [44,45]. Under acidic conditions (pH < 5.82), the amino group in the TEPA@ATA@AC composite was protonated, thus showing a positive charge, which could repel cationic Pb(II) ions, leading to a declined uptake efficacy and uptake capacity. As the pH level increases, the protonation degree of active constituents diminishes. Consequently, incorporation through ion exchange, electrostatic attraction, and chelation between functional groups on the TEPA@ATA@AC composite and Pb(II) ions became more robust enhancing adsorption capacity. After pH 6.5, Lead(II) hydroxide precipitates as PbOH⁺, Pb(OH)₂, and Pb(OH)₃⁻^1^ [46,47,48]. Therefore, a pH of 6.5 was designated as the optimal adsorbent pH for the subsequent tests.

#### 2.2.3. Effect of Composite Dose

Various adsorbent dosages were evaluated to determine how they affect the adsorption process (from 0.005 to 0.05 g), as presented in Figure 5c. The findings revealed that the uptake efficiency of Pb(II) improved from 57.3% to 98.0%, with an increase in the TEPA@ATA@AC dosage from 0.005 to 0.02. Then it was almost stable as the adsorbent mass increased to 0.05 g, while the elimination capacity of TEPA@ATA@AC toward Pb(II) declined from 114.6 mg/g to 19.7 mg/g as the TEPA@ATA@AC dosage was boosted from 0.005 g to 0.05 g. The improvement in the uptake efficacy of Pb(II) was related to a rise in the number of uptake sites and the surface area of TEPA@ATA@AC [36]. Therefore, 0.02 g of TEPA@ATA@AC was chosen as the optimal mass for further measurements.

### 2.3. Modeling Adsorption

#### 2.3.1. Isotherms

Various Pb(II) concentrations were assessed to determine how they affect adsorption (from 20 to 300 mg/L at pH = 6.5, T = 298–318 K, agitation = 100 rpm), as depicted in Figure 6a. The findings revealed that the uptake capacity of TEPA@ATA@AC improved with Pb(II) from 48.68 mg/g to 410.4 mg/g, with a rise in the original Pb(II) concentration from 20 mg/L to 300 mg/L. At 100 and 150 mg/L the removal efficiency was 91.56% and 89.34%, respectively. Then, the greater the original Pb(II) ion contents, the slower the uptake rate was owing to a saturation of the active sites on the TEPA@ATA@AC with Pb(II) ions [49]. The uptake of Pb(II) on the TEPA@ATA@AC composite was also analyzed at different temperatures (298, 308, 318 K) under the same experimental condition. According to Figure 6a, by changing the temperature from 298 K to 318 K, a tiny reduction in the elimination capacity from 48.68 mg/g to 42.55 mg/g at 20 mg/L and from 410.4 mg/g to 324.25 mg/g at 300 mg/L indicated the uptake of Pb(II) on the TEPA@ATA@AC composite was exothermic. This further reflects that the uptake of Pb((II) by the TEPA@ATA@AC composite was caused by physisorption coupled with limited chemisorption [50]. Based on these results, a temperature of 298 K was selected for further measurements.

To explore the uptake mechanism of Pb(II) ions on the TEPA@ATA@AC composite and determine the maximum adsorption capacity, the adsorption findings were tested using three nonlinear isotherm models; the Langmuir [51], Freundlich [52], and Dubinin–Radushkevich models [53] were investigated. Table 2 provides the variables of isotherm models and the fit information is exhibited in Figure 6a–c. Based on the regression factor (R^2^), as presented in Table 2, the Langmuir model fit the data better (0.95435 ≤ R^2^ ≤ 0.97283) than the Freundlich (0.87741≤ R^2^ ≤ 0.92251) and the Dubinin–Radushkevich models (0.88103 ≤ R^2^ ≤ 0.9017), which indicated that the uptake of Pb(II) onto the TEPA@ATA@AC composite was primarily through homogeneous and monolayer adsorption. The highest adsorption capacity of the TEPA@ATA@AC composite evaluated using the Langmuir model was 432.8 mg/g at 298 K, which was higher than previously functionalized activated carbon adsorbents for uptake of Pb(II), as shown in Table 3 [4,27,36,54,55,56,57,58,59,60,61,62,63]. Furthermore, the decrease in KL values from 0.1407 to 0.0488 L/mg as the temperature rose from 298 to 318 K suggests that the TEPA@ATA@AC composite exhibited a higher affinity for Pb(II) at 298 K [64]. The n values in the Freundlich model were in the range of 2.99 < n < 3.79, indicating a favorable adsorption process. According to the Dubinin–Radushkevich model (Table 4), the mean energy E at different temperatures (298–318 K) was in the range (kJ/mol). Since all these values were below 8.0 kJ/mol, it indicates that the uptake of Pb^2+^ was primarily governed by physical adsorption [64].

#### 2.3.2. Adsorption Kinetics

To select the minimum contact time for the elimination of Pb(II) ions using the TEPA@ATA@AC composite, a range of time intervals was examined from 5 to 480 min at [Pb(II)]_0_ = 20 mg/L, pH = 6.5, T = 298 K, and agitation = 100 rpm, as depicted in Figure 6d. According to this figure, the Pb(II) removal efficiency was rapid, especially in the first 60 min, where an uptake efficacy of 73.45% was obtained. Then, the uptake performance was gradually increased with an increase in the uptake time until the equilibrium was maintained at 300 min with a maximum uptake efficiency of 97.65%. In addition, the maximum adsorption capacity of TEPA@ATA@AC toward Pb(II) was 48.83 mg/g at 300 min. The rapid uptake of Pb(II) within the first 60 min could be due to the abundance of accessible active uptake centers of the TEPA@ATA@AC surface [3,65]. After the equilibrium period (300 min), there was no remarkable alteration in elimination activity because of the complete covering of the adsorption centers on the TEPA@ATA@AC surface. Hence, a duration of 300 min was determined as the optimal equilibrium time for subsequent measurements.

The adsorption data were examined utilizing various kinetics models such as Elovich, pseudo-first-order (PFO), and pseudo-second-order (PSO) models to emphasize the uptake mechanism of Pb(II) ions onto TEPA@ATA@AC composite. Table 4 provides the variables of the kinetics models and the plot of the nonlinear kinetics is exhibited in Figure 6d. As per the regression factor coefficients (R^2^), as depicted in Table 3, the R^2^ magnitude for the PFO, PSO, and Elovich models was 0.99335, 0.99086, and 0.95356, respectively. This indicates that the PFO and PSO models, with R2 values greater than 0.99, provided a better fit to the data compared to the Elovich model. This suggests that the uptake of Pb(II) by the TEPA@ATA@AC composite was primarily driven by physisorption, with limited involvement of chemisorption [50]. The provided finding indicates a sophisticated uptake process that encompasses a combination of pseudo-first-order and pseudo-second-order kinetics, in addition to fitting the Langmuir isotherm. The pseudo-first-order kinetics model, assuming a direct proportionality between the rate of adsorption and the unoccupied sites on the adsorbent surface, often presents itself in a linear equation.

On the other hand, the pseudo-second-order kinetics model, which suggests a chemisorption mechanism by asserting that the uptake rate is proportional to the square of the unoccupied sites, is typically represented as a linear equation. The Langmuir isotherm, a model describing solute adsorption onto an adsorbent with a finite number of identical sites, assumes monolayer adsorption without interaction between adsorbed molecules. In addition, the q_e_. _exp_ value obtained from the experimental data of 48.85 mg/g was closer to the PFO model value of 48.13 mg/g, and closer to the q_e_. _cal_ value obtained from the PFO model (48.13 mg/g) than those in the PSO model (53.47 mg/g), implying implementing of the PFO model to kinetic data.

#### 2.3.3. Thermodynamic Parameters

Thermodynamic variables, such as Δ*G*°, Δ*S*°, and Δ*H*°, were determined to emphasize the character of the uptake process. These variables were estimated using the following Equations (1)–(3):(1)$${lnK}_{e}^{0}=-\frac{\Delta H{}^{\circ}}{RT}+\frac{\Delta S{}^{\circ}}{R}$$
(2)$$∆G{}^{\circ}=-RT\;lnK_{e}^{0}$$
(3)$$K_{e}^{0}=\frac{(1000*K_{L}*molecular\;weight\;of\;adsorbate)\cdot [adsrobate]{}^{\circ}}{\gamma }$$
where $K_{e}^{0}$ (L·mol^−1^) is an equilibrium constant, *K_L_* (L·mg^−1^) is the Langmuir constant, *R* is the gas constant (8.314 J K^−1^ mol^−1^), and γ is the activity coefficient [66,67,68]. The values of Δ*H*° and Δ*S*° were calculated from the plot of ln $K_{e}^{0}$ versus 1/*T*. The thermodynamic variables at various temperatures (298, 308, and 318 K) are presented in Table 5. The negative magnitude of Δ*G*° showed that the Pb(II) adsorption on the TEPA@ATA@AC composite was spontaneous. In addition, the Δ*G*° increased with increasing temperature, suggesting that the high uptake efficiency of Pb(II) occurred at low temperatures (298 K). The negative enthalpy (Δ*H*°) was −42.16 kJ/mol. This reflects that the interaction of Pb(II) uptake by TEPA@ATA@AC was exothermic. The negative Δ*S*° value implies a decline in randomness during the adsorption of Pb(II).

### 2.4. Removal Mechanism

The notable efficiency in Pb(II) uptake (97.7%) and the high capacity (432.8 mg/g) of Pb(II) on the TEPA@ATA@AC composite due to the abundant presence of amino groups (-NH_2_, -NH), along with -OH and -COOH moieties, on the surface of TEPA@ATA@AC is illustrated in Figure 7. The results obtained from the FTIR and EDX analysis before and after adsorption affirmed the successful Pb(II) adsorption onto the TEPA@ATA@AC composite. The FTIR analysis revealed that the change in the intensities of bands and disappearances of bands could be attributed to the elimination of Pb(II) onto the surface of the TEPA@ATA@AC composite. In detail, the band at 3341 cm^−1^ and (1581, 1471 cm^−1^) for –OH/–NH stretching and –NH_2_ bending experienced a slight decrease in intensity and a minor shift towards a higher wavenumber of 3345 cm^−1^ and (1583, 1474 cm^−1^), respectively, indicating the binding of Pb(II) and Pb(OH)^+^ with functional groups on TEPA@ATA@AC through electrostatic interaction. The bands C-N and C-O disappeared after Pb(II) adsorption. All the above changes indicate that the hydroxyl, carboxyl, and primary or secondary amino moieties of TEPA@ATA@AC participated in the adsorption of Pb(II). The results from the pH effect revealed that the uptake of Pb(II) on the TEPA@ATA@AC composite depends on the pH value. As the pH increased, the active groups’ protonation decreased, strengthening the columbic interaction between the functional moieties of TEPA@ATA@AC and Pb(II) ions, increasing the uptake capacity. Based on the kinetics and isotherm models, the uptake of Pb(II) by the TEPA@ATA@AC composite can be attributed to physisorption (electrostatic interaction) coupled with limited chemisorption (coordination interactions).

### 2.5. Regeneration of the Adsorbent

The desorption efficiency of Pb(II) from the TEPA@ATA@AC composite was carried out using the three eluents HNO_3_(0.1 M), HCl (0.1 M), and CH_3_COOH (0.1 M), as shown in Figure 8a. According to the results of the effect of pH, the Pb(II) removal efficiency was very low (3.95%) in an acidic medium (pH 2.6); therefore, three types of acidic reagents were applied to the desorption of Pb(II) from the adsorbent surface. Lu et al. studied the desorption of Pb(II) from vanadium, titanium-bearing magnetite coated with humic acid (VTM-HA) in the pH range 1–7. They found that the Pb(II) desorption reached 99.3 wt% at pH 1 [69]. From Figure 7, 0.1 M HNO_3_ had the greatest desorption efficiency of 91.14%, followed by 0.1 M HCl, which exhibited a desorption efficiency of 89.09%, and the 0.1 M CH_3_COOH had the minimum of 70.56%%. The high desorption of Pb(II) with 0.1 M HNO_3_ was due to the large number of H^+^ ions that can replace the Pb(II) ions adsorbed on the surface of the adsorbent [70]. Similar findings were observed for the elimination of Pb(II) ions employing EGTA–chitosan by Zhao et al. They found that 2 M HNO_3_ was a good reagent for the reusability of Pb(II) ions [71]. Five runs of successive adsorption/desorption tests were conducted utilizing 0.1 M HNO_3_ as eluent, as depicted in Figure 8b. From the first to the fifth cycle, the desorption efficiency of Pb(II) from the TEPA@ATA@AC composite reduced from 91.14% to 79.18% and the uptake efficacy of Pb(II) declined from 97.65% to 83.80%. This may be due to the loss of adsorbent caused by the experiment or the incomplete desorption. Similar findings were observed on the elimination of Pb(II) ions employing *Polygonum orientale*-activated carbon by Zhang et al. They found that 0.1 M HCl was a good reagent for the reusability of Pb(II) ions [72].

## 3. Experimental

### 3.1. Materials

Tetraethylenepentamine (≥97%), 1-Ethyl-3-(3 dimethylaminopropyl) carbodiimide (EDC), and 2-aminoterephthalic acid (ATA, ≥99%) were brought from Sigma-Aldrich (St. Louis, MO, USA). Dimethyl formamide (DMF, 99.8%) was provided from Panreac, Barcelona, Apain. Sodium hydroxide (98%) was obtained from BDH, Poole, UK. Hydrochloric acid (37%), nitric acid (70%), and ammonium hydroxide (25%) were obtained from Merck, Darmstadt, Germany. Acetic acid (CH_3_COOH) was obtained from Qualichem, Verginia, USA. The production of activated carbon was accomplished through chemical activation of date palm pits (Phoenix dactylifera seeds) using a method pioneered by Aldawsari et al. [45].

### 3.2. Characterization

The surface morphology of TEPA@ATA@AC composite was examined prior and post Pb(II) uptake using scanning electron microscope (JEOL-JSM 6380, Los Angeles, CA, USA). The elemental assessment was also assessed using energy-dispersive X-ray (AMETEK Nova 200, Berwyn, PA, USA). FT-IR spectroscopy (Thermo Scientific, Nicolet 6700, Waltham, Massachusetts, USA) in the 400–4000 cm^−1^ was employed to examine the function moieties of the prepared composites. The thermal durability of the TEPA@ ATA@AC nanocomposite was detected by TGA (Mettler Toledo, GA/SDTA851, Columbus, OH, USA). The point Zeta potential of TEPA@ATA@AC was measuring utilizing a Zeta potential analyzer (Nano Plus Series, Particulate Systems, York, PA, USA). The crystalline structure AC, ATA@AC, and TEPA@ATA@AC was recorded on X-ray diffraction (XRD: Shimadzu model 6000 Corporation, Kyoto, Japan).

### 3.3. Preparation of TEPA@ATA@AC Composite

Tetraethylenepentamine-grafted amino terephthalic acid-functionalized activated carbon (TEPA@ATA@AC) was fabricated according to the following procedure: First, acid-modified activated carbon was synthesized via some modification of the previously reported approach [73]. Activated carbon (AC) underwent a series of modifications to yield ATA@AC. Initially, 3 g of AC was immersed in 250 mL of a solution of HNO_3_, followed by magnetic stirring for 4 h at 50 °C. The resulting AC-COOH was separated, rinsed with DI water, and subsequently dried in an oven overnight at 80 °C. Next, AC-COOH was further modified with amino terephthalic acid to produce ATA@AC. This involved introducing 1.068 g of EDC to a solution containing 1.271 g of ATA in DMF (50 mL). The blend was agitated magnetically for 30 min, after which 1.8 g of AC-COOH was added and stirred for an additional 24 h. The resultant ATA@AC was filtered, rinsed with DMF, ethanol, and DI water, then dried in an oven at 50 °C for 24 h. Finally, TEPA@ATA@AC was prepared through the in situ integration of TEPA with the COOH moiety of ATA@AC using EDC as a coupling agent. Specifically, 5 mL of TEPA was introduced to 1.2 g of ATA@AC suspended in a 30 mL DMF solution, along with 0.941 g of EDC, using ultrasonication for 30 min at room temperature. The suspension was then subjected to magnetic stirring for 24 h at 50 °C. The obtained TEPA@ATA@AC was isolated, rinsed with DMF, ethanol, and DI water, and dried in an oven overnight at 50 °C. The entire fabrication process is illustrated in Figure 9.

### 3.4. Adsorption Tests

The impact of different parameters such as contact time (5–480 min), initial Pb(II) concentration (20–300 mg/L) and pH (2.6–7), dosage (0.005–0.05 g), and temperature (298–308 K) on Pb(II) adsorption were studied using batch adsorption experiments. In general, 50 mL of Pb(II) ion solution (20 ppm (mg/L)) was mixed with 0.02 g of TEPA@ATA@AC compound. Then, the pH of the solution was monitored utilizing HNO_3_ and NaOH solutions. After that, the blend was stirred at a rate of 100 rpm for 300 min. After adsorption, the blend was separated and the filtrate was collected to measure the contents of the Pb(II) ions utilizing AAS. Finally, the uptake capacity *q_e_* (mg/g) of TEPA@ATA@AC composite toward Pb(II) and the uptake efficacy R(%) were estimated using Equations (4) and (5), respectively.
(4)$$q_{e}(mg/g)=(C_{o}-C_{e})\frac{V}{m}$$
(5)$$R(\%)=\frac{C_{o}-C_{e}}{C_{o}}\times 100$$
where *C_o_* and *C_e_* (mg/L) are the original Pb(II) ions and equilibrium concentrations of Pb(II), respectively; V is the volume; and m (g) is the quantity of TEPA@ATA@AC adsorbent.

For the desorption study, the initial step involving the Pb(II) uptake of TEPA@ATA@AC was performed under the optimum conditions. After uptake, the Pb(II) adsorbed on TEPA@ATA@AC was regenerated using two eluents (0.1 M HCl and 0.1 M HNO_3_) in the following manner: The Pb(II) adsorbed on TEPA@ATA@AC adsorbent was rinsed with deionized water, then introduced into a conical flask containing 50 mL of appropriate eluent. After shaking for 300 min, the sample was separated and the quantity of Pb(II) in solution was determined utilizing AAS. The desorption of Pb(II) dye was estimated applying Equation (6):(6)$$\%Desorption=\frac{C_{m}}{C_{e}}\times 100$$
where *C_m_* (mg/L) is the content of Pb(II) desorbed and *C_e_* (mg/L) is the concentration of Pb(II) adsorbed.

## 4. Conclusions

In this study, tetraethylenepentamine (TEPA) was integrated into amino terephthalic acid-modified activated carbon (ATA@AC) through in situ coupling using EDC as a coupling agent. This resulted in the formation of the TEPA@ATA@AC composite, designed for the elimination of Pb(II). A characterization via SEM, EDX, FT-IR, TGA, XRD, and Zeta potential confirmed its composition and structure. The Pb(II) removal efficacy of the TEPA@ATA@AC composite was exceptional, achieving a 97.65% removal rate under optimal conditions (pH = 6.5, contact time = 300 min, adsorbent dosage = 0.02 g, and temperature = 298 K). Kinetic studies suggested that the uptake process fitted the PFO and PSO models, suggesting a combination of physisorption and limited chemisorption. The equilibrium data were well affirmed by the Langmuir isotherm, with a greatest monolayer adsorption capacity of 432.8 mg/g at 298 K. Thermodynamic assessments revealed that the adsorption process was exothermic and feasible. Additionally, the TEPA@ATA@AC composite displayed promising regeneration and reusability capabilities.

## Figures and Tables

**Figure 1 molecules-29-01586-f001:**
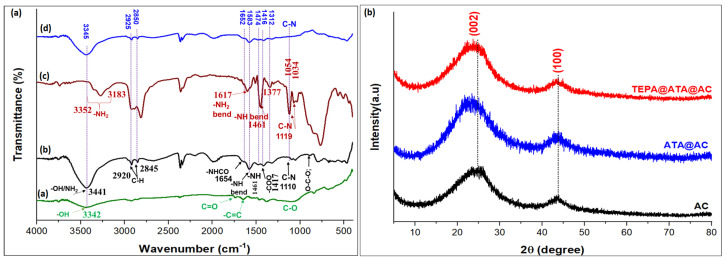
FTIR spectrum of AC (line a), TEPA@ATA@AC (line b), TEPA (line c), and TEPA@ATA@AC/Pb(II) (line d) (**a**); and XRD patterns of the AC, ATA@AC, and TEPA@ATA@AC composite (**b**).

**Figure 2 molecules-29-01586-f002:**
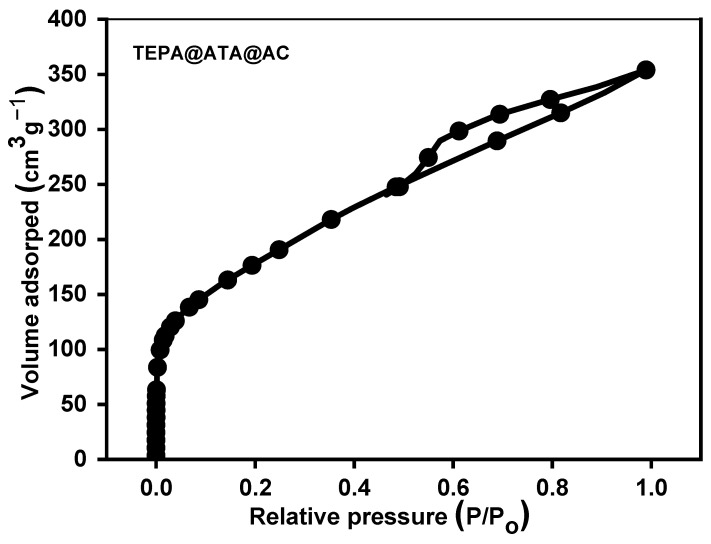
Nitrogen isotherm of TEPA@ATA@AC sample.

**Figure 3 molecules-29-01586-f003:**
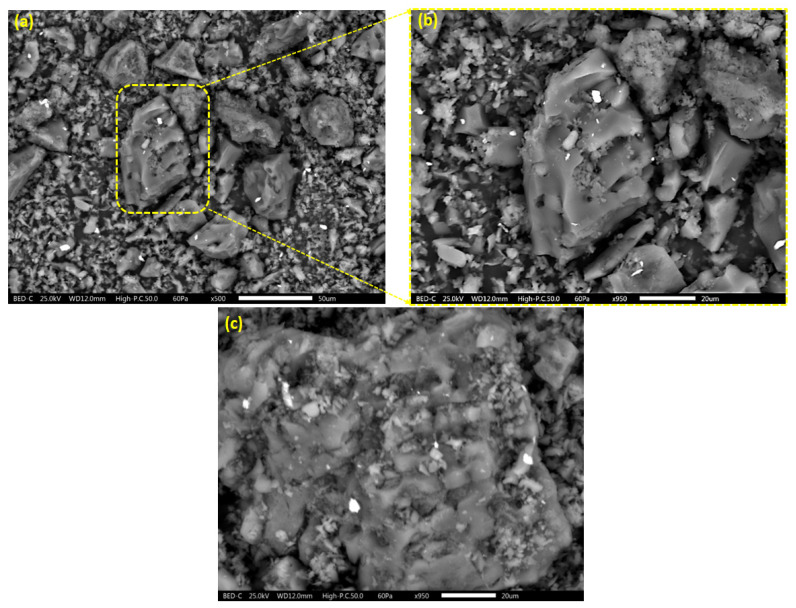
SEM images of TEPA@ATA@AC (**a**,**b**) and TEPA@ATA@AC after Pb(II) adsorption (**c**).

**Figure 4 molecules-29-01586-f004:**
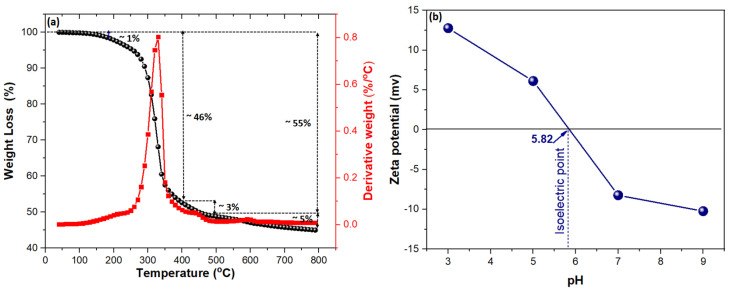
TGA/DTA curves (**a**) and Zeta potential to TEPA@ATA@AC composite (**b**).

**Figure 5 molecules-29-01586-f005:**
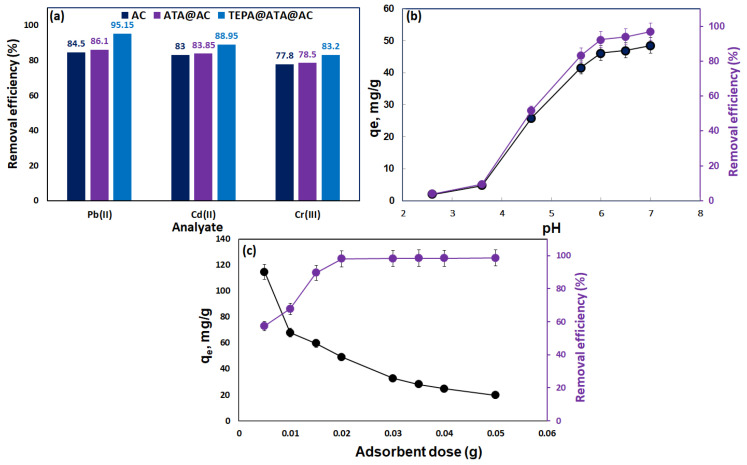
Selectivity study (condition: [metal]: 20 mg/L; time: 24 h; adsorbent mass: 0.02 g; temperature: 298 K; pH: 6) (**a**), effect of initial pH (Condition: [Pb(II)] = 20 mg/L, adsorbent mass = 0.02 g, T = 298 K, time = 24 h, agitation = 100 rpm) (**b**), adsorbent mass (condition: [Pb(II)] = 20 mg/L, T = 298 K, pH: 6.5, time = 24 h, agitation = 100 rpm) on adsorption of Pb(II) on TEPA@ATA@AC composite (**c**).

**Figure 6 molecules-29-01586-f006:**
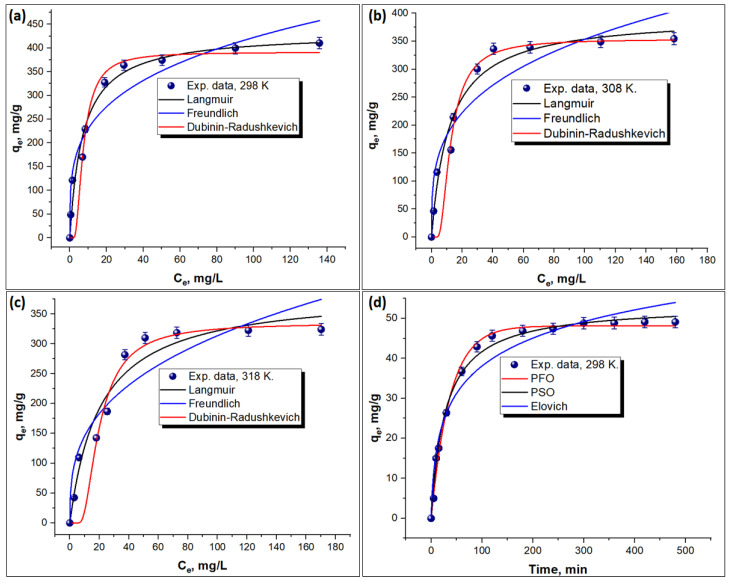
Adsorption isotherms for Pb(II) on TEPA@ATA@AC composite at 298 K (**a**), 308 K (**b**), and 318 K (**c**); adsorption kinetic for Pb(II) on TEPA@ATA@AC composite at 298 K (**d**).

**Figure 7 molecules-29-01586-f007:**
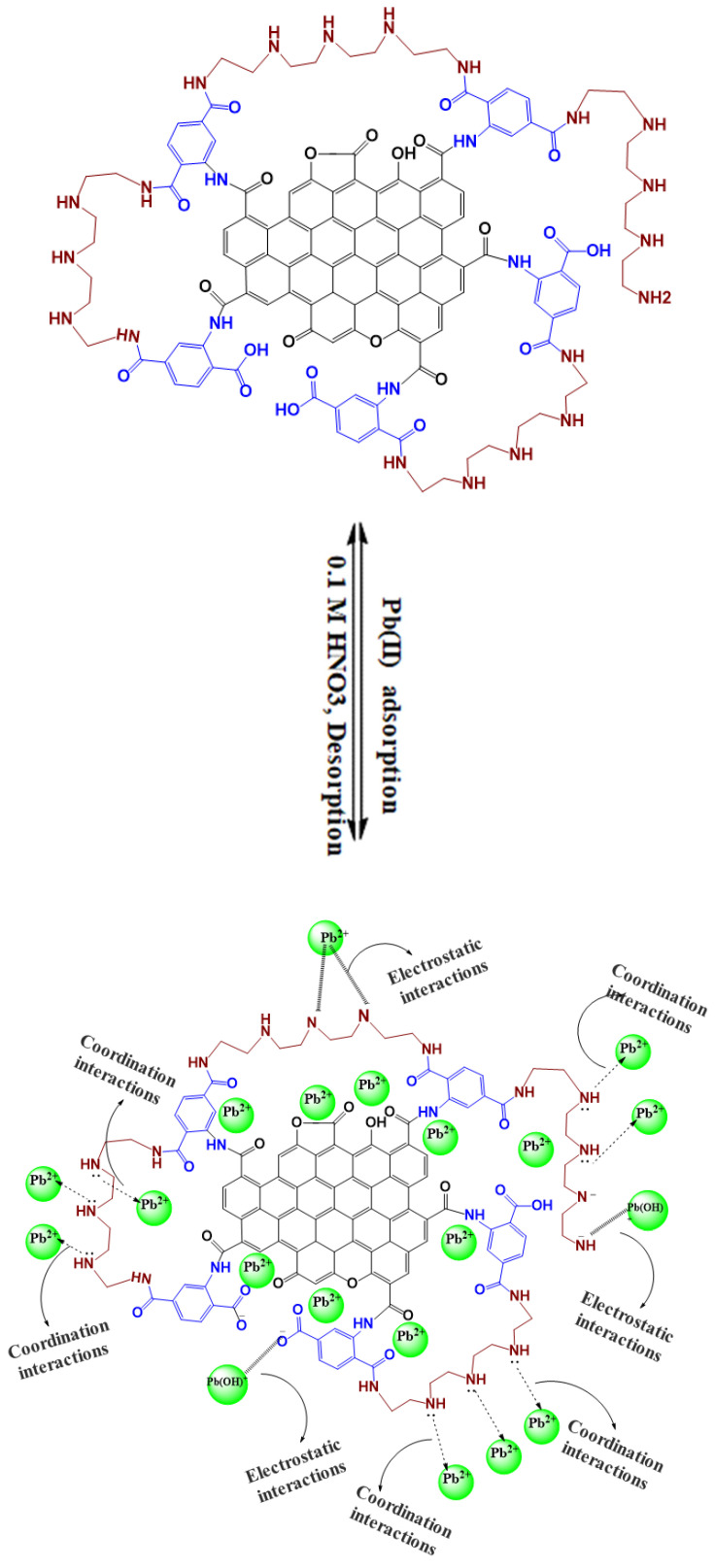
Mechanism adsorption of Pb(II) on TEPA@ATA@AC composite.

**Figure 8 molecules-29-01586-f008:**
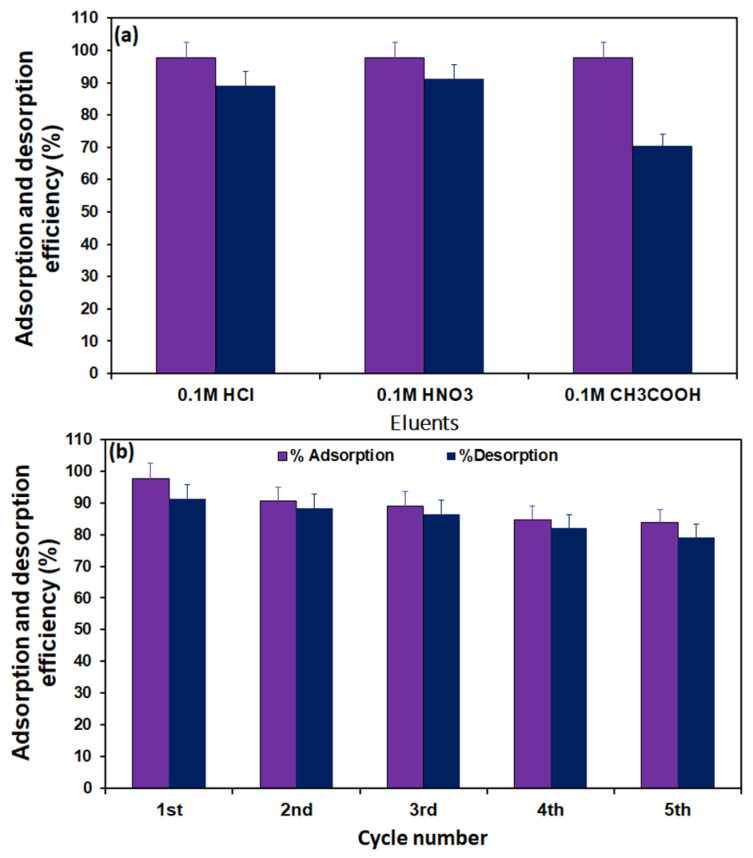
Desorption plots of Pb(II) from saturated TEPA@ATA@AC composite (**a**), and recycling of TEPA@ATA@AC composite for the removal of Pb(II) (**b**).

**Figure 9 molecules-29-01586-f009:**
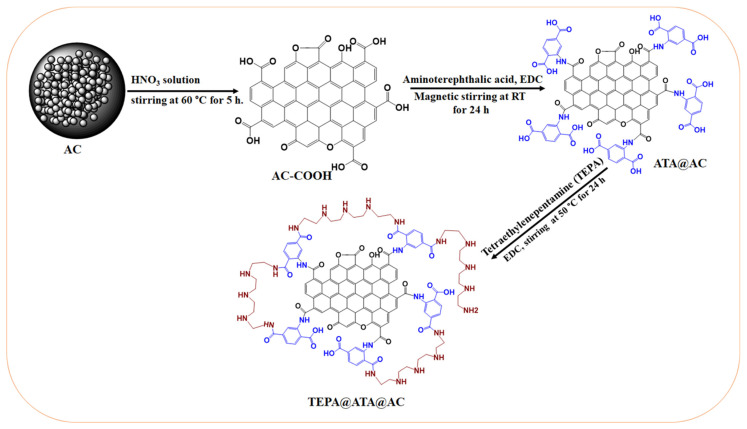
Scheme illustrating the preparation of TEPA@ATA@AC composite.

**Table 1 molecules-29-01586-t001:** EDX analysis.

Sample	Elemental Content (Wt.%)			
	C	O	N	Pb(II)
TEPA@ATA@AC	72.33	24.20	3.47	-
TEPA@ATA@AC/Pb(II)	73.42	22.78	2.92	0.88

**Table 2 molecules-29-01586-t002:** Isotherm data for uptake of Pb(II) on TEPA@ATA@AC.

Adsorbent	T (K)	*q_e,exp_.* (mg/g)	*Langmuir*			*Freundlich*				*Dubinin-R*		
			*q_m_*, (mg/g)	*K_L_* (L/mg)	*R^2^*	*K_F_* (mg^1−1/n^·L^1/n^ g^−1^)	n	*R* ^2^	*q_s_*, mg/g	*K_D-R_* (mol^2^ KJ^−2^)	*E* (kJ mol^−1^)	*R* ^2^
TEPA@ATA@AC	298	410.4	432.8	0.1407	0.9728	125.4	3.79	0.9225	391.3	18.4	2.24	0.9017
	318	354.5	395.7	0.0843	0.9687	93.5	3.46	0.8925	353.9	49.5	2.89	0.8874
	328	324.3	387.8	0.0488	0.9543	67.3	2.99	0.8774	334.8	117.5	10	0.8810

**Table 3 molecules-29-01586-t003:** Comparison of the Pb(II) adsorption capacity of on TEPA@ATA@AC composite with other modified activated carbon adsorbents.

Adsorbent	Conditions	Adsorption Capacity (mg/g)	Ref.
Sulfhydryl functionalized activated carbon	[Pb(II)]_0_—20–70 mg/L; pH—5.6; T—298 K; dose—20 mg; volume—0.02 L; time—120 min	116.3	[4]
Thiol-functionalized activated carbon	[Pb(II)]_0_—5–300 mg/L; pH—5; T—298 K; dose—20 mg; volume—0.02 L; time—1440 min	232.02	[36]
Cobalt ferrite-supported activated carbon	[Pb(II)]_0_—0.849 mg/L; pH—5; T—333 K; dose—800 mg; volume—0.1 L; time—80 min	6.27	[54]
Sodium alginate (TSA) and activated carbon fiber	[Pb(II)]_0_—400 mg/L; pH—5; T—303 K; dose—30 mg; volume—0.1 L; time—30 min	221.3	[55]
Eriochrome Blue Black-modified activated carbon	[Pb(II)]_0_—150 mg/L; pH—7; T—298 K; dose—25 mg; volume—0.025 L; time—60 min	127.9	[56]
Magnetized activated carbons (MAC)	[Pb(II)]_0_—40–700 mg/L; pH—5; T—298 K; dose—0.25 mg/L: time—720 min	253.2	[57]
AC/Fe_3_O_4_@SiO_2_–NH_2_	[Pb(II)]_0_—20–250 mg/L; pH—5.2; T—303 K; dose—20 mg; volume—0.025 L; time—1440 min	104.2	[58]
Bamboo-activated carbon@SiO_2_-ETDA	[Pb(II)]_0_—50–100 mg/L; pH—5.3; T—303 K; dose—20 mg; volume—0.025 L; time—1440 min.	45.5	[59]
Humic acid modified activated carbon adsorbent (AC-HA)	[Pb(II)]_0_—50–100 mg/L; pH—5.5; T—303 K; dose—30 mg; volume—0.05 L; time—720 min	250	[60]
2-Aminothiazol-modified activated carbon (AT-AMC)	[Pb(II)]_0_—10–800 mg/L; pH—5.5; T—298 K; dose—10 mg; volume—0.01 L; time—60 min	310.9	[61]
Polyethylenimine-modified carbon thin film (ACTF-PEI)	[Pb(II)]_0_—5–250 mg/L; pH—5; T—298 K; dose—20 mg; volume—0.02 L; time—30 min	143	[62]
Magnetic-activated carbon incorporated with amino groups	[Pb(II)]_0_—5–250 mg/L; pH—6; T—298.15 K; dose—10 mg; volume—0.02 L; time—180 min	104.2	[63]
TEPA@ATA@AC	[Pb(II)]_0_—20–300 mg/L; pH—6.5; T—298 K; dose—20 mg; volume—0.05 L; time—300 min	432.8	This study

**Table 4 molecules-29-01586-t004:** Kinetic data for Pb(II) ions on TEPA@ATA@AC.

[Pb(II)]_0_ (mg/L)	*q_e,exp_*. (mg/g)	*Pseudo-First-Order*			*Pseudo-Second-Order*			*Elovich*		
		*q_e, cal_.* (mg/g)	*K*_1_ (1/min)	*R^2^*	*q_e2, cal_.* (mg/g)	*K*_2_ (g/mg-min)	*R* ^2^	*A* (mg/g min)	*Β* (mg/g)	*R* ^2^
20	48.85	48.13	0.027	0.9933	53.47	6.61.10–4	0.9929	4.09	0.097	0.9536

**Table 5 molecules-29-01586-t005:** Thermodynamic parameters for Pb(II) adsorption on TEPA@ATA@AC.

Adsorbent	Temperature (K)	Δ*G* (kJ mol^−1^)	Δ*H* (kJ mol^−1^)	Δ*S* (J mol^−1^ K^−1^)
TEPA@ATA@AC	298	−23.95	−42.16	−60.99
	318	−23.45		
	328	−22.73		

## Data Availability

The data presented in this study are available on request from the corresponding authors.

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
