# Peer review of "Tetraethylenepentamine-Grafted Amino Terephthalic Acid-Modified Activated Carbon as a Novel Adsorbent for Efficient Removal of Toxic Pb(II) from Water"

_molecules, 2024, doi:10.3390/molecules29071586_

Round 1

Reviewer 1 Report

Comments and Suggestions for Authors

The authors addressed all comments sufficiently. 

Comments on the Quality of English Language

English need some editing

Author Response

Reviewer 1

Comment 1: Comments and Suggestions for Authors

The authors addressed all comments sufficiently. 

Response: Dear reviewer, I appreciate you for your precious time in reviewing my article.

Comment 2: Comments on the Quality of English Language

English need some editing

Response: Thank you so much for your nice comment. I have revised the English language and highlighted it in blue.

Reviewer 2 Report

Comments and Suggestions for Authors

This manuscript meets the standard recommended by the journal. However, before its acceptance in this journal, a minor revision is suggested. My comments are as follows:

-The studied material was characterized quite poorly in the manuscript. This needs to be supplemented. Raman spectra and characteristics of the porous structure (nitrogen adsorption isotherms, porous structure parameters and pore size distribution) would be valuable.

-The content of functional groups on the surface of the tested materials plays a large role in the Pb2+ sorption process. This type of research needs to be completed.

- Digits should be corrected on subscripts on lines 539, 572,590, 601.

- In Table 2, page 10, the barracks of the KF unit.

- All References should follow the Molecules rules.

Author Response

Reviewer 2

Comments and Suggestions for Authors

This manuscript meets the standard recommended by the journal. However, before its acceptance in this journal, a minor revision is suggested. My comments are as follows:

Dear reviewer, I appreciate you for your precious time in reviewing my article and look forward to meeting your expectations. Below is my point-by-point response to your valuable comments and suggestions.

Comment 1: The studied material was characterized quite poorly in the manuscript. This needs to be supplemented. Raman spectra and characteristics of the porous structure (nitrogen adsorption isotherms, porous structure parameters and pore size distribution) would be valuable.

Response: Dear reviewer, thank you sincerely for your positive feedback. In this study, I conducted a comprehensive characterization of the adsorbent using various techniques, including SEM, EDX, FT-IR, TGA, XRD, and Zeta potential. Unfortunately, our university currently does not have access to Raman spectroscopy equipment. Nevertheless, I believe that the FTIR analysis provides sufficient insights into the adsorbent's properties. Regarding the BET surface area analysis, the Central Laboratory is currently undergoing technical maintenance, which is taking longer than expected due to international import shortages. I assure you that your valuable comment will be taken into consideration in our future work. Thank you for your understanding.

Comment 2: The content of functional groups on the surface of the tested materials plays a large role in the Pb2+ sorption process. This type of research needs to be completed.

Response: Dear reviewer, I appreciate your positive feedback. I conducted EDX analysis to examine the elemental composition of the TEPA@ATA@AC composite. The results revealed the presence of carbon (72.33%), oxygen (24.20%), and nitrogen (3.47%) elements on the surface of the TEPA@ATA@AC composite. This observation confirms the abundance of functional groups on the composite surface. Your comments are valuable, and we will take them into consideration in our future work.

Comment 3: Digits should be corrected on subscripts on lines 539, 572,590, 601.

Response: Thank you for your kind comment. I have corrected the subscript digits for atoms in lines 539, 572, 590, and 601 in the reference sections.

Comment 4: In Table 2, page 10, the barracks of the KF unit.

Response: Thank you for your update. I have added the missing unit of Freundlich constant KF in the Table 2 as KF (mg1-1/n. L1/n g-1).

Comment 5: All References should follow the Molecules rules.

Response: Thank you for your kind comment. I have reviewed and adjusted all reference formats in the entire manuscript to align with the guidelines of the Journal of Molecules.

Reviewer 3 Report

Comments and Suggestions for Authors

The submitted manuscript is a classical work reporting the synthesis of a novel activated carbon modified with amine groups to enhance the ability to effectively remove Pb(II) from water. The work covers the synthesis and characterization of the material, a screening test in liquid phase with three heavy metals and liquid phase assays to study the kinetic and equilibrium removal of Pb(II), thermodynamics, and regeneration.

The major fragility of the manuscript is the fact that only tests the novel activated carbon developed for the target application with to sequential functionalization steps (TEPA@APA@AC) lacking the direct comparison with the unmodified activated carbon (AC) or the intermediate material only functionalized with the amino terephthalic acid (ATA@AC). Thus not allowing the direct comparison and consequently a deeper understanding of the advantages of the two-step modification process for this specific application. Besides, the lack of information to understand the advantages of the functionalization and extra cost of the new material compared with the AC, its practical application is questionable since the selected dosage (0.02g per 50 mL of solution) is not feasible in a real scale application.

The major problems that must be solved before considering publication:

1.      Lines 85-89: if for Cd removal thiol group is better than the amino group why the author only explores amine-functionalization? Or at least why not to address both?

2.      Topic 3.1 for direct comparison FTIR of ATA@AC and AC must be presented

3.      An activated carbon (AC) and derived materials (ATA@AC and TEPA@APA@AC) must always be characterized by N2 adsorption at -196 ºC in order to obtain information regarding the micro/mesopore structure its most distinctive and unique features

4.      SEM micrographies do not allow to characterize the nanoporous structure (apertures bellow 50 nm) of an activated carbon or derived material – line 208

5.      Lines 211-218: when presenting SEM-EDX data of heretogeneous materials as those presented in Figure 3 the author most clarify the region analised. What is the elemental composition of the bigger particles with a dense structure and some channels? What is the elemental composition of the smaller and reticulated particles in Fig 3a? Ideally present the mappings by element with the corresponding SEM image. Is TEPA really attached to the AC?

6.      Line 232 the Fe3O4@CNC-TEPA is not the material developed or studied in this manuscript, correct.

7.      For the direct comparison and consequently a deeper understanding of the advantages of the two-step modification process to obtain TEPA@APA@AC for this specific application the characterization and liquid assays must also be performed and presented for APA@AC and AC.

8.      Integrate the data of kinetic and equilibrium assays along the topics to avoid repetition. No need to present graphics from Fig. 5d and 5e since Fig 6 present the same experimental points with the fitting to the most appropriate models.

9.      Line 291 correct “adsorption capacity” since for pH > 6.5 occurs precipitation thus can not be considered that occurs adsorption.

10.   The comparison of Table 3 must be done with cares since in most cases distinct experimental conditions were used and thus leading to different adsorption capacities

11.   Topic 3.3.2 expressions “bi-pore structure” or “micropores” cannot be used since not N2 adsorption was presented.

Formatting, references and minor points:

1.      Uniform the references along the text:

a.      most of times [X] but other times “name at al (year)” – eg. Line 69

b.      line 197 it is [4, 35] and not [35], [4] – correct all equivalents along the manuscript

2.      Line 98 – adsorb and not “absorb”

3.      ATA missing in topic 2.1

4.      Explain the experimental protocol to obtain the graphic that allows to determine the zeta potential

5.      Explain why the selection of two acids at specific concentrations for the desorption study

6.      Correct typos along the manuscript, eg. Adsorption (Table 3), Adsorbate (line 421)…

7.      Avoid the exact same phase in different parts of the document, eg. Line 399 was already used, lines 441-445 repeat info previously presented

8.      Explain the activity coefficient (line 423)

9.      Line 487 mentions CH3COOH that was not reported in experimental section

Author Response

Reviewer 3

Comments and Suggestions for Authors

The submitted manuscript is a classical work reporting the synthesis of a novel activated carbon modified with amine groups to enhance the ability to effectively remove Pb(II) from water. The work covers the synthesis and characterization of the material, a screening test in liquid phase with three heavy metals and liquid phase assays to study the kinetic and equilibrium removal of Pb(II), thermodynamics, and regeneration.

The major fragility of the manuscript is the fact that only tests the novel activated carbon developed for the target application with to sequential functionalization steps (TEPA@APA@AC) lacking the direct comparison with the unmodified activated carbon (AC) or the intermediate material only functionalized with the amino terephthalic acid (ATA@AC). Thus not allowing the direct comparison and consequently a deeper understanding of the advantages of the two-step modification process for this specific application. Besides, the lack of information to understand the advantages of the functionalization and extra cost of the new material compared with the AC, its practical application is questionable since the selected dosage (0.02g per 50 mL of solution) is not feasible in a real scale application.

Response: I would like to thank you for taking the necessary time and effort to review the manuscript. I sincerely appreciate all your valuable comments and suggestions, which helped us in improving the quality of the manuscript. Those comments are all valuable and very helpful for revising and improving the paper.  I have studied the comments carefully and have made corrections which we hope meet you with approval. Revised sections are marked in blue in the revised paper. The following is a point-to-point response to your comments.

The major problems that must be solved before considering publication:

Comment 1: Lines 85-89: if for Cd removal thiol group is better than the amino group why the author only explores amine-functionalization? Or at least why not to address both?

Response: Dear reviewer, in the introduction, we included some literature about using tetraethylenepentamine (TEPA) to remove heavy metals from aqueous solutions. The authors prepared two adsorbents, amino-functionalized activated carbon (N-AC) and thiol-functionalized activated carbon (S-AC) by stepwise modification with tetraethylenepentamine (TEPA), cyanuric chloride (CC), and sodium sulfide. They concluded that the (S-AC) showed a good removal for Pb(II) and Cd(II) compared to (N-AC). The second adsorbent (TEPA-GO@MnFe2O4) showed a higher adsorption capacity of TEPA-GO@MnFe2O4 toward Pb(II) (263 mg/g) compared to GO@MnFe2O4 (133 mg/g) and original GO (196 mg/g). [27]. I modified the surface of ATA@AC with tetraethylenepentamine (TEPA) due to the amine groups having an affinity toward heavy metals. Amino functional groups can strongly adsorb heavy metal ions, thus improving the removal efficiency. in this study, the results exhibited that the TEPA@ATA@AC composite exhibited superior Pb(II) removal performance with high removal efficiency (97.65%) and uptake capacity (432.8 mg/g).

Comment 2: Topic 3.1 for direct comparison FTIR of ATA@AC and AC must be presented

Response: thank you so much for your nice comment. As per your suggestion, I have added the FTIR for AC in the Fig. 2 and text as " The FTIR spectra of AC (line a) showed the broad band at 3441 cm-1, indicating the presence of–OH stretching of phenolic, alcohol, and carboxylic acid compounds. The bands at 2844 – 2914 cm-1(C-H aliphatic stretching), 1733 (-COO- stretch), 1635 cm-1 (aromatic -C=C stretching vibrations), 1377- 1464 cm-1 (C-H aliphatic bending). The band at 1071 cm-1 and 831 cm-1 are attributed to the ν(C-O bond)[39–41]. "

Comment 3: An activated carbon (AC) and derived materials (ATA@AC and TEPA@APA@AC) must always be characterized by N2 adsorption at -196 ºC in order to obtain information regarding the micro/mesopore structure its most distinctive and unique features.

Response: Dear reviewer, thank you so much for your nice comment. In this study, I characterized the adsorbent by different techniques, including SEM, EDX, FT-IR, TGA, XRD, and Zeta potential. Regarding the BET surface area, at present, there is some technical maintenance in the Central Laboratory, which takes a longer time due to the shortage in international imports. Dear reviewer, this comment will be considered in our future work.

Comment 4: SEM micrographies do not allow to characterize the nanoporous structure (apertures bellow 50 nm) of an activated carbon or derived material – line 208

Response: Dear reviewer, thank you so much for your nice comment. According to SEM analysis, it was observed some porous in the surface of TEPA@ATA@AC composite. But as per your suggestion, I have remove the sentence in line 208

Comment 5: Lines 211-218: when presenting SEM-EDX data of heretogeneous materials as those presented in Figure 3 the author most clarifies the region analised. What is the elemental composition of the bigger particles with a dense structure and some channels? What is the elemental composition of the smaller and reticulated particles in Fig 3a? Ideally present the mappings by element with the corresponding SEM image. Is TEPA really attached to the AC?

Response: Dear reviewer, thank you so much for your nice comment. Regarding the mappings of elements for the TEPA@ATA@AC composite, I did not get mapping images with results of SEM images from the central Laboratory. However, the formation of TEPA@ATA@AC composite was successfully confirmed by FTIR analysis. Dear reviewer, this comment will be considered in our future work.

Comment 6: Line 232 the Fe3O4@CNC-TEPA is not the material developed or studied in this manuscript, correct.

Response: Thank you so much. I have corrected this typos error.

Comment 7: For the direct comparison and consequently a deeper understanding of the advantages of the two-step modification process to obtain TEPA@ATA@AC for this specific application the characterization and liquid assays must also be performed and presented for ATA@AC and AC.

Response: Thank you for your thoughtful review of our manuscript. We appreciate your feedback and the opportunity to address your concerns. Regarding the characterization, we acknowledge the importance of providing a comprehensive comparison between ATA@AC and AC to deepen the understanding of the advantages of the two-step modification process for obtaining TEPA@ATA@AC for our specific application. We would like to assure you that we have already performed XRD for both ATA@AC and AC, and the results have been included in the manuscript.

Comment 8: Integrate the data of kinetic and equilibrium assays along the topics to avoid repetition. No need to present graphics from Fig. 5d and 5e since Fig 6 present the same experimental points with the fitting to the most appropriate models.

Response: Thank you so much. As per your suggestion. I have removed Fig. 5d and 5 e and moved the effect of the contact time section (3.2.4) and the effect of the initial Pb(II) concentration section (3.2.5) to the kinetics section (3.3.1) and isotherm section (3.3.2), respectively.

Comment 9: Line 291 correct “adsorption capacity” since for pH > 6.5 occurs precipitation thus can not be considered that occurs adsorption.

Response: Thank you so much. I have corrected.

Comment 10: The comparison of Table 3 must be done with cares since in most cases distinct experimental conditions were used and thus leading to different adsorption capacities

Response: Thank you very much. According to your suggestion. I have added to Table 3 a comparison in terms of experimental conditions such as contact time, adsorbent dose, initial concentration, pH, and temperature.

Comment 11: Topic 3.3.2 expressions “bi-pore structure” or “micropores” cannot be used since not N2 adsorption was presented.

Response: Thank you so much. As per your suggestion. I have corrected.

Formatting, references and minor points:

Comment 1: Uniform the references along the text

 Response: Thank you so much. I have uniformed the references along the text.

Comment a: most of times [X] but other times “name at al (year)” – eg. Line 69

Response: Thank you very much. Throughout the text, I use [ ] as a cited reference, and for “name et al” I only mention the author who did the work.

– eg. Line 69 " Amine-modified corn husk-derived activated carbon was prepared by Ismail et al. for the elimination of Ni(II), Cu(II), and Pb (II) from battery recycling contaminated water. They found that the highest uptake capacity of Ni(II), Cu(II), and Pb(II) ions was 0.337 0.724, and 2.814 mg/g, respectively [22]."

Comment b: line 197 it is [4, 35] and not [35], [4] – correct all equivalents along the manuscript

Response: Thank you so much. I have corrected the references along the manuscript

Comment 2: Line 98 – adsorb and not “absorb”

Response:  Thank you so much. I have corrected this typos error.

Comment 3: ATA missing in topic 2.1

Response: Thank you so much. I have added the ATA to the topic 2.1 as 2-aminoterephthalic acid (ATA, ≥99%). 

Comment 4: Explain the experimental protocol to obtain the graphic that allows to determine the zeta potential.

Response: Thank you so much. The zeta potential was measured as a function of pH at room temperature. For this purpose, the TEPA@TAT@AC composite was suspended in deionized water and then sonicated for 10 min. The pHs of the dispersions were adjusted to the desired values with HCl (0.1M) or NaOH (0.1 M). The dispersions were then allowed to settle for 24 h and the supernatant was used for zeta-potential measurement.

Comment 5: Explain why the selection of two acids at specific concentrations for the desorption study

Response: Thank you so much. Based on the effect of pH on the removal efficiency of Pb(II), the removal efficiency of Pb(II) in acidic medium (pH= 2.6) is very small (3.95% ), therefore, I used three eluents (HNO3(0.1 M), HCl (0.1 M), and CH3COOH (0.1M), ) and the result showed that 0.1 M HNO3 had the greatest desorption efficiency of 91.14%. Many previous studies have indicated that acid eluent is an effective eluent for the desorption of lead from the adsorbent surface [Journal of Hazardous Materials 167 (2009) 1242–1245], [Carbohydrate Polymers 80 (2010) 891–899], [Chemical Engineering Journal 220 (2013) 412–419]

 Comment 6: Correct typos along the manuscript, eg. Adsorption (Table 3), Adsorbate (line 421).

Response: Corrected

 Comment 7: Avoid the exact same phase in different parts of the document, eg. Line 399 was already used, lines 441-445 repeat info previously presented

Response: Thank you so much. The whole manuscript was rechecked and revised.

Comment 8: Explain the activity coefficient (line 423)

The activity of the adsorbate is defined as:  ???????? ????????? = [?????????]. γ ,

where [Adsorbate] is the molar concentration of the adsorbate (mol L-1); and γ is the coefficient of activity of the adsorbate (dimensionless). The activity coefficient is unitary for diluted solutions [Journal of Molecular Liquids 352 (2022) 118762], [J Mol Liq. 273 (2019) 425–434.]

Comment 9: Line 487 mentions CH3COOH that was not reported in experimental section

Response: Thank you so much. I have added the CH3COOH in the experimental section. 

Round 2

Reviewer 3 Report

Comments and Suggestions for Authors

The revised manuscript replies only to some of the points highlighted during the review, not solving the two major fragilities identified in the first review report:

-        absence of textural characterization by N2 adsorption at -196 ºC due to maintenance of the lab (previous comment 3)

-        no presentation of liquid phase results regarding AC or ATA@AC to evaluate the advantage of the two-step modification process on the improvement of the adsorption capacity for Pb(II)

Regarding the second point the 433 mg/g adsorption capacity attained with the TEPA@ATA@AC is higher than the values reported in the literature proving the high adsorption capacity of the material but lacking the information to support the role of the two modifications (ATA and TEPA) on the enhancement of the adsorption capacity of the as prepared AC.

Besides these major comments the other point still no be solved are the following:

New reviewer comment: The name of the 2-aminoterephthalic acid is defined as ATA in line 108 and most of times is presented as so, but along the manuscript also appears as TAT (e.g. line 116, 127). Correct and uniform nomenclature along all the text

Previous Comment 4: SEM micrographies do not allow to characterize the nanoporous structure (apertures bellow 50 nm) of an activated carbon or derived material – line 208

Response: Dear reviewer, thank you so much for your nice comment. According to SEM analysis, it was observed some porous in the surface of TEPA@ATA@AC composite. But as per your suggestion, I have remove the sentence in line 208

Reviewer reply: The sentence was not removed (now in line 218) and in figure 3b channels are identified as pores. Since no textural characterization was made and AC are nanoporous materials the word “pores” cannot be used in this context, since being AC nanoporous materials  word “porous” can be misleadingly considered the nanopores of the materials.

Previous Comment 5: Lines 211-218: when presenting SEM-EDX data of heretogeneous materials as those presented in Figure 3 the author most clarifies the region analised. What is the elemental composition of the bigger particles with a dense structure and some channels? What is the elemental composition of the smaller and reticulated particles in Fig 3a? Ideally present the mappings by element with the corresponding SEM image. Is TEPA really attached to the AC?

Response: Dear reviewer, thank you so much for your nice comment. Regarding the mappings of elements for the TEPA@ATA@AC composite, I did not get mapping images with results of SEM images from the central Laboratory. However, the formation of TEPA@ATA@AC composite was successfully confirmed by FTIR analysis. Dear reviewer, this comment will be considered in our future work.

Reviewer reply: Not done limiting the comprehension of the EDX data presented.

Previous Comment 10: The comparison of Table 3 must be done with cares since in most cases distinct experimental conditions were used and thus leading to different adsorption capacities

Response: Thank you very much. According to your suggestion. I have added to Table 3 a comparison in terms of experimental conditions such as contact time, adsorbent dose, initial concentration, pH, and temperature.

Reviewer reply: Recommend to use “[Pb(II)]0” instead of “Co” for initial concentration of Pb(II) and regarding adsorbent dose it must be presented in mg/L or mass of adsorbent and volume of solution, otherwise the comparison cannot be made

Previous minor Comment 5: Explain why the selection of two acids at specific concentrations for the desorption study

Response: Thank you so much. Based on the effect of pH on the removal efficiency of Pb(II), the removal efficiency of Pb(II) in acidic medium (pH= 2.6) is very small (3.95% ), therefore, I used three eluents (HNO3(0.1 M), HCl (0.1 M), and CH3COOH (0.1M), ) and the result showed that 0.1 M HNO3 had the greatest desorption efficiency of 91.14%. Many previous studies have indicated that acid eluent is an effective eluent for the desorption of lead from the adsorbent surface [Journal of Hazardous Materials 167 (2009) 1242–1245], [Carbohydrate Polymers 80 (2010) 891–899], [Chemical Engineering Journal 220 (2013) 412–419]

Reviewer reply: These literature studies must be cited and discussed in the manuscript

Author Response

Comments and Suggestions for Authors

The revised manuscript replies only to some of the points highlighted during the review, not solving the two major fragilities identified in the first review report:

Comment 1-   absence of textural characterization by N2 adsorption at -196 ºC due to maintenance of the lab (previous comment 3)

Response: The nitrogen isotherm of the prepared sample (TEPA@ATA@AC) was provided in the revised version (See Figure 3)

Comment 2-   no presentation of liquid phase results regarding AC or ATA@AC to evaluate the advantage of the two-step modification process on the improvement of the adsorption capacity for Pb(II)

Response: Dear reviewer, thank you so much for your nice comment. After preparing the adsorbents, I studied the selectivity of three adsorbents (AC, ATA@AC, and TEPA@ATA@AC) toward Pb(II), Cd(II), and Cr(III), and the results are shown in Section (3.2.1. Selectivity Study (Fig. 5a)). Also, I have added the result regarding the AC in the text of the selectivity study section as "the revealed that the TEPA@ATA@AC showed a higher removal efficiency for Cd(II), Pb(II), and Cr(III) metals in the following rank: Pb(II) (95.15%)>Cd(II)(88.95%)>Cr(III)(83.20%) compared to the AC and ATA@AC composite which have the adsorption order:  Pb(II)(84.50%)>Cd(II)(83%)>Cr(III)(77.80%) for AC adsorbent and Pb(II)(86.10%)>Cd(II)(83.85%)> Cr(III)(78.50%) for ATA@AC composite".

Comment 3-   Regarding the second point the 433 mg/g adsorption capacity attained with the TEPA@ATA@AC is higher than the values reported in the literature proving the high adsorption capacity of the material but lacking the information to support the role of the two modifications (ATA and TEPA) on the enhancement of the adsorption capacity of the as prepared AC.

Response: Dear reviewer, thank you so much. I explained the reasons that led to the enhancement of the adsorption capacity of TEPA@ATA@AC toward Pb(II) in the text (section 3.2.1) as ". This is attributed to the structure of the TEPA@ATA@AC composite containing multifunctional groups (NH2, OH, and COOH) compared to the AC and ATA@AC composite, which contain only COOH and OH groups. The formation of TEPA@ATA@AC composite was confirmed by FTIR analysis.

Comment 4-   Besides these major comments the other point still no be solved are the following:

New reviewer comment: The name of the 2-aminoterephthalic acid is defined as ATA in line 108 and most of times is presented as so, but along the manuscript also appears as TAT (e.g. line 116, 127). Correct and uniform nomenclature along all the text

Response: Thank you so much nice comment. I have corrected this typo error. Now, the name of the adsorbent is TEPA@ATA@AC in all of the text.

Comment 5-   Previous Comment 4: SEM micrographies do not allow to characterize the nanoporous structure (apertures bellow 50 nm) of an activated carbon or derived material – line 208

Response1: Dear reviewer, thank you so much for your nice comment. According to SEM analysis, it was observed some porous in the surface of TEPA@ATA@AC composite. But as per your suggestion, I have remove the sentence in line 208

Reviewer reply: The sentence was not removed (now in line 218) and in figure 3b channels are identified as pores. Since no textural characterization was made and AC are nanoporous materials the word “pores” cannot be used in this context, since being AC nanoporous materials word “porous” can be misleadingly considered the nanopores of the materials.

Comments and Suggestions for Authors

The revised manuscript replies only to some of the points highlighted during the review, not solving the two major fragilities identified in the first review report:

Comment 1-   absence of textural characterization by N2 adsorption at -196 ºC due to maintenance of the lab (previous comment 3)

Response: The nitrogen isotherm of the prepared sample (TEPA@ATA@AC) was provided in the revised version (See Figure 3)

Comment 2-   no presentation of liquid phase results regarding AC or ATA@AC to evaluate the advantage of the two-step modification process on the improvement of the adsorption capacity for Pb(II)

Response: Dear reviewer, thank you so much for your nice comment. After preparing the adsorbents, I studied the selectivity of three adsorbents (AC, ATA@AC, and TEPA@ATA@AC) toward Pb(II), Cd(II), and Cr(III), and the results are shown in Section (3.2.1. Selectivity Study (Fig. 5a)). Also, I have added the result regarding the AC in the text of the selectivity study section as "the revealed that the TEPA@ATA@AC showed a higher removal efficiency for Cd(II), Pb(II), and Cr(III) metals in the following rank: Pb(II) (95.15%)>Cd(II)(88.95%)>Cr(III)(83.20%) compared to the AC and ATA@AC composite which have the adsorption order:  Pb(II)(84.50%)>Cd(II)(83%)>Cr(III)(77.80%) for AC adsorbent and Pb(II)(86.10%)>Cd(II)(83.85%)> Cr(III)(78.50%) for ATA@AC composite".

Comment 3-   Regarding the second point the 433 mg/g adsorption capacity attained with the TEPA@ATA@AC is higher than the values reported in the literature proving the high adsorption capacity of the material but lacking the information to support the role of the two modifications (ATA and TEPA) on the enhancement of the adsorption capacity of the as prepared AC.

Response: Dear reviewer, thank you so much. I explained the reasons that led to the enhancement of the adsorption capacity of TEPA@ATA@AC toward Pb(II) in the text (section 3.2.1) as ". This is attributed to the structure of the TEPA@ATA@AC composite containing multifunctional groups (NH2, OH, and COOH) compared to the AC and ATA@AC composite, which contain only COOH and OH groups. The formation of TEPA@ATA@AC composite was confirmed by FTIR analysis.

Comment 4-   Besides these major comments the other point still no be solved are the following:

New reviewer comment: The name of the 2-aminoterephthalic acid is defined as ATA in line 108 and most of times is presented as so, but along the manuscript also appears as TAT (e.g. line 116, 127). Correct and uniform nomenclature along all the text

Response: Thank you so much nice comment. I have corrected this typo error. Now, the name of the adsorbent is TEPA@ATA@AC in all of the text.

Comment 5-   Previous Comment 4: SEM micrographies do not allow to characterize the nanoporous structure (apertures bellow 50 nm) of an activated carbon or derived material – line 208

Response1: Dear reviewer, thank you so much for your nice comment. According to SEM analysis, it was observed some porous in the surface of TEPA@ATA@AC composite. But as per your suggestion, I have remove the sentence in line 208

Reviewer reply: The sentence was not removed (now in line 218) and in figure 3b channels are identified as pores. Since no textural characterization was made and AC are nanoporous materials the word “pores” cannot be used in this context, since being AC nanoporous materials word “porous” can be misleadingly considered the nanopores of the materials.

Comments and Suggestions for Authors

The revised manuscript replies only to some of the points highlighted during the review, not solving the two major fragilities identified in the first review report:

Comment 1-   absence of textural characterization by N2 adsorption at -196 ºC due to maintenance of the lab (previous comment 3)

Response: The nitrogen isotherm of the prepared sample (TEPA@ATA@AC) was provided in the revised version (See Figure 3)

Comment 2-   no presentation of liquid phase results regarding AC or ATA@AC to evaluate the advantage of the two-step modification process on the improvement of the adsorption capacity for Pb(II)

Response: Dear reviewer, thank you so much for your nice comment. After preparing the adsorbents, I studied the selectivity of three adsorbents (AC, ATA@AC, and TEPA@ATA@AC) toward Pb(II), Cd(II), and Cr(III), and the results are shown in Section (3.2.1. Selectivity Study (Fig. 5a)). Also, I have added the result regarding the AC in the text of the selectivity study section as "the revealed that the TEPA@ATA@AC showed a higher removal efficiency for Cd(II), Pb(II), and Cr(III) metals in the following rank: Pb(II) (95.15%)>Cd(II)(88.95%)>Cr(III)(83.20%) compared to the AC and ATA@AC composite which have the adsorption order:  Pb(II)(84.50%)>Cd(II)(83%)>Cr(III)(77.80%) for AC adsorbent and Pb(II)(86.10%)>Cd(II)(83.85%)> Cr(III)(78.50%) for ATA@AC composite".

Comment 3-   Regarding the second point the 433 mg/g adsorption capacity attained with the TEPA@ATA@AC is higher than the values reported in the literature proving the high adsorption capacity of the material but lacking the information to support the role of the two modifications (ATA and TEPA) on the enhancement of the adsorption capacity of the as prepared AC.

Response: Dear reviewer, thank you so much. I explained the reasons that led to the enhancement of the adsorption capacity of TEPA@ATA@AC toward Pb(II) in the text (section 3.2.1) as ". This is attributed to the structure of the TEPA@ATA@AC composite containing multifunctional groups (NH2, OH, and COOH) compared to the AC and ATA@AC composite, which contain only COOH and OH groups. The formation of TEPA@ATA@AC composite was confirmed by FTIR analysis.

Comment 4-   Besides these major comments the other point still no be solved are the following:

New reviewer comment: The name of the 2-aminoterephthalic acid is defined as ATA in line 108 and most of times is presented as so, but along the manuscript also appears as TAT (e.g. line 116, 127). Correct and uniform nomenclature along all the text

Response: Thank you so much nice comment. I have corrected this typo error. Now, the name of the adsorbent is TEPA@ATA@AC in all of the text.

Comment 5-   Previous Comment 4: SEM micrographies do not allow to characterize the nanoporous structure (apertures bellow 50 nm) of an activated carbon or derived material – line 208

Response1: Dear reviewer, thank you so much for your nice comment. According to SEM analysis, it was observed some porous in the surface of TEPA@ATA@AC composite. But as per your suggestion, I have remove the sentence in line 208

Reviewer reply: The sentence was not removed (now in line 218) and in figure 3b channels are identified as pores. Since no textural characterization was made and AC are nanoporous materials the word “pores” cannot be used in this context, since being AC nanoporous materials word “porous” can be misleadingly considered the nanopores of the materials.

Comments and Suggestions for Authors

The revised manuscript replies only to some of the points highlighted during the review, not solving the two major fragilities identified in the first review report:

Comment 1-   absence of textural characterization by N2 adsorption at -196 ºC due to maintenance of the lab (previous comment 3)

Response: The nitrogen isotherm of the prepared sample (TEPA@ATA@AC) was provided in the revised version (See Figure 3)

Comment 2-   no presentation of liquid phase results regarding AC or ATA@AC to evaluate the advantage of the two-step modification process on the improvement of the adsorption capacity for Pb(II)

Response: Dear reviewer, thank you so much for your nice comment. After preparing the adsorbents, I studied the selectivity of three adsorbents (AC, ATA@AC, and TEPA@ATA@AC) toward Pb(II), Cd(II), and Cr(III), and the results are shown in Section (3.2.1. Selectivity Study (Fig. 5a)). Also, I have added the result regarding the AC in the text of the selectivity study section as "the revealed that the TEPA@ATA@AC showed a higher removal efficiency for Cd(II), Pb(II), and Cr(III) metals in the following rank: Pb(II) (95.15%)>Cd(II)(88.95%)>Cr(III)(83.20%) compared to the AC and ATA@AC composite which have the adsorption order:  Pb(II)(84.50%)>Cd(II)(83%)>Cr(III)(77.80%) for AC adsorbent and Pb(II)(86.10%)>Cd(II)(83.85%)> Cr(III)(78.50%) for ATA@AC composite".

Comment 3-   Regarding the second point the 433 mg/g adsorption capacity attained with the TEPA@ATA@AC is higher than the values reported in the literature proving the high adsorption capacity of the material but lacking the information to support the role of the two modifications (ATA and TEPA) on the enhancement of the adsorption capacity of the as prepared AC.

Response: Dear reviewer, thank you so much. I explained the reasons that led to the enhancement of the adsorption capacity of TEPA@ATA@AC toward Pb(II) in the text (section 3.2.1) as ". This is attributed to the structure of the TEPA@ATA@AC composite containing multifunctional groups (NH2, OH, and COOH) compared to the AC and ATA@AC composite, which contain only COOH and OH groups. The formation of TEPA@ATA@AC composite was confirmed by FTIR analysis.

Comment 4-   Besides these major comments the other point still no be solved are the following:

New reviewer comment: The name of the 2-aminoterephthalic acid is defined as ATA in line 108 and most of times is presented as so, but along the manuscript also appears as TAT (e.g. line 116, 127). Correct and uniform nomenclature along all the text

Response: Thank you so much nice comment. I have corrected this typo error. Now, the name of the adsorbent is TEPA@ATA@AC in all of the text.

Comment 5-   Previous Comment 4: SEM micrographies do not allow to characterize the nanoporous structure (apertures bellow 50 nm) of an activated carbon or derived material – line 208

Response1: Dear reviewer, thank you so much for your nice comment. According to SEM analysis, it was observed some porous in the surface of TEPA@ATA@AC composite. But as per your suggestion, I have remove the sentence in line 208

Reviewer reply: The sentence was not removed (now in line 218) and in figure 3b channels are identified as pores. Since no textural characterization was made and AC are nanoporous materials the word “pores” cannot be used in this context, since being AC nanoporous materials word “porous” can be misleadingly considered the nanopores of the materials.

Comments and Suggestions for Authors

The revised manuscript replies only to some of the points highlighted during the review, not solving the two major fragilities identified in the first review report:

Comment 1-   absence of textural characterization by N2 adsorption at -196 ºC due to maintenance of the lab (previous comment 3)

Response: The nitrogen isotherm of the prepared sample (TEPA@ATA@AC) was provided in the revised version (See Figure 3)

Comment 2-   no presentation of liquid phase results regarding AC or ATA@AC to evaluate the advantage of the two-step modification process on the improvement of the adsorption capacity for Pb(II)

Response: Dear reviewer, thank you so much for your nice comment. After preparing the adsorbents, I studied the selectivity of three adsorbents (AC, ATA@AC, and TEPA@ATA@AC) toward Pb(II), Cd(II), and Cr(III), and the results are shown in Section (3.2.1. Selectivity Study (Fig. 5a)). Also, I have added the result regarding the AC in the text of the selectivity study section as "the revealed that the TEPA@ATA@AC showed a higher removal efficiency for Cd(II), Pb(II), and Cr(III) metals in the following rank: Pb(II) (95.15%)>Cd(II)(88.95%)>Cr(III)(83.20%) compared to the AC and ATA@AC composite which have the adsorption order:  Pb(II)(84.50%)>Cd(II)(83%)>Cr(III)(77.80%) for AC adsorbent and Pb(II)(86.10%)>Cd(II)(83.85%)> Cr(III)(78.50%) for ATA@AC composite".

Comment 3-   Regarding the second point the 433 mg/g adsorption capacity attained with the TEPA@ATA@AC is higher than the values reported in the literature proving the high adsorption capacity of the material but lacking the information to support the role of the two modifications (ATA and TEPA) on the enhancement of the adsorption capacity of the as prepared AC.

Response: Dear reviewer, thank you so much. I explained the reasons that led to the enhancement of the adsorption capacity of TEPA@ATA@AC toward Pb(II) in the text (section 3.2.1) as ". This is attributed to the structure of the TEPA@ATA@AC composite containing multifunctional groups (NH2, OH, and COOH) compared to the AC and ATA@AC composite, which contain only COOH and OH groups. The formation of TEPA@ATA@AC composite was confirmed by FTIR analysis.

Comment 4-   Besides these major comments the other point still no be solved are the following:

New reviewer comment: The name of the 2-aminoterephthalic acid is defined as ATA in line 108 and most of times is presented as so, but along the manuscript also appears as TAT (e.g. line 116, 127). Correct and uniform nomenclature along all the text

Response: Thank you so much nice comment. I have corrected this typo error. Now, the name of the adsorbent is TEPA@ATA@AC in all of the text.

Comment 5-   Previous Comment 4: SEM micrographies do not allow to characterize the nanoporous structure (apertures bellow 50 nm) of an activated carbon or derived material – line 208

Response1: Dear reviewer, thank you so much for your nice comment. According to SEM analysis, it was observed some porous in the surface of TEPA@ATA@AC composite. But as per your suggestion, I have remove the sentence in line 208

Reviewer reply: The sentence was not removed (now in line 218) and in figure 3b channels are identified as pores. Since no textural characterization was made and AC are nanoporous materials the word “pores” cannot be used in this context, since being AC nanoporous materials word “porous” can be misleadingly considered the nanopores of the materials.vv
